# Mechanistic and Functional Studies on the Microbial Induction of *Wolfiporia cocos* Liquid Fermentation Products

**DOI:** 10.3390/foods13101578

**Published:** 2024-05-18

**Authors:** Zhikang Yang, Congbao Su, Zhoujie Xu, Yiting Liu, Jianhui Chen, Xiaoping Wu

**Affiliations:** College of Life Sciences, Fujian Agriculture and Forestry University, Fuzhou 350002, China; zhikangyang5851@163.com (Z.Y.); 17720815813@163.com (C.S.); xzj54772024@163.com (Z.X.); lyt991022@163.com (Y.L.); 15059064565@163.com (J.C.)

**Keywords:** microbial induction, co-fermentation, intracellular polysaccharide, antitumor, transcriptomics

## Abstract

Liquid fermentation is an efficient culture for obtaining polysaccharides from edible mushrooms. In this study, the polysaccharide content and biomass were examined by introducing microorganisms into the *Wolfiporia cocos* fermentation system. Three edible mushroom co-fermentation systems were established, among which the *Wolfiporia cocos-Ganoderma lucidum* co-fermentation system significantly increased the mycelial biomass of the system by 57.71% compared to *Wolfiporia cocos* alone and 91.22% compared to *Ganoderma lucidum* alone, and the intracellular polysaccharide content was significantly increased. Physiological activities of polysaccharides showed that mycelial polysaccharides in the *Wolfiporia cocos-Ganoderma lucidum* system had stronger anti-tumor cell value-adding and anti-tumor cell migration activities compared with *Wolfiporia cocos* and *Ganoderma lucidum* fermentation alone. The transcriptomic study of *Wolfiporia cocos* mycelium induced by exogenous substances suggested that the exogenous substances could enhance the intracellular polysaccharide content of *Wolfiporia cocos* through the upregulation of the expression of α-glycosyltransferase encoded by ALG10 and the downregulation of α-glycosidases encoded by MAN1B in the glycolytic metabolism of *Wolfiporia cocos*. This study provides a new direction for the transformation of polysaccharides from *Wolfiporia cocos* and *Ganoderma lucidum* into functional foods and new product development, and provides an experimental basis.

## 1. Introduction

One of the primary active ingredients of *Wolfiporia cocos* (*Wolfiporia cocos* (Schwein.) Ryvarden & Gilb.), a large member of the Polyporaceae family of fungi, is *Wolfiporia* polysaccharide. It is primarily found in the roots of Pinus pinnatifida plants, and the dried fungal nuclei are utilized for medicinal purposes [1]. Polysaccharides are predominantly formed through the dehydration and condensation of glycosidic linkages and are generated from monosaccharides, such as glucose [2], xylose, rhamnose, mannose and galactose. The primary component of these polymers is β-(1→3)-D-glucan [3,4,5]. *W. cocos*, renowned for its efficacy in alleviating water seepage dampness in medicine, is beneficial for individuals with modern lifestyles, including those who stay up late and experience symptoms of dampness due to irregular diets. Due to its distinct whitening properties, it is extensively used in healthy food and cosmetic industries [6,7,8]. The demand is increasing every year, and it is difficult to meet the market demand by relying on wild products and artificial bionic cultivation [9]. Thus, the key to solve this major contradiction is to find a fast and effective method of *Wolfiporia cocos* culture. At this stage, most *Wolfiporia cocos* polysaccharides come from wild *Wolfiporia cocos*, fermented *Wolfiporia cocos*, lime wood *Wolfiporia cocos* and bagged *Wolfiporia cocos* [10]. *Wolfiporia cocos* polysaccharides can be obtained via the liquid fermentation method with less time and space resources than other culture methods, which is a hot research topic at present. Fermentation environment, medium composition and strain activity are three important factors affecting the level of liquid fermentation. Zhang Shiquan [11] et al. Three single factors, inoculum, incubation temperature and incubation time in *Wolfiporia cocos* liquid culture conditions, were optimized to improve *Wolfiporia cocos* polysaccharide yield. The nutrient matrix is mixed with an appropriate amount of specially selected Chinese herbal medicines containing active ingredients as the medicinal matrix, and the product is transformed from medicinal mycoplasm to medicinal mycoplasm with corresponding medicinal substances. This increases fermentation levels and enhances drug activity and efficacy [12]. Wang biao [13] et al. A high-yield GABA fermentation system for erythromycin-fermented mulberry leaves was obtained by optimizing fermentation parameters using artificial neural network. Zhang [14] et al. The biotransformation of *Ganoderma lucidum* was stimulated by the glycoside components in the Tongkuan vine to prepare a product rich in stimulating saponins. The anticancer effect was significantly enhanced, mainly by inhibiting the growth and apoptosis of cancer cells. Liu Jia [15] et al. Kombucha was fermented using *Phellinus igniarius* and the physiological functions of the products were tested. The results showed a significant increase in anti-tumor ability, presumably an effect brought about by the increase in polysaccharides and total phenolic substances of the product. However, there was no significant effect on some cancer cells [16]. The addition of exogenous substances to the fermentation system to promote fermentation is widely used in the study of medicinal fungi fermentation, and it is of great significance for the development and application of medicinal fungi resources to improve the fermentation level of medicinal fungi [17]. Most of the previous studies established plant and fungal co-fermentation systems in order to explore a co-fermentation system established by a variety of fungi. The anti-tumor ability was used as an indicator to find out the specific molecular mechanisms affecting the change of active ingredients in the co-fermentation system. Since there are few studies on liquid co-fermentation [18,19,20], with our findings we aim to provide reference value for a wide range of applications in the field of functional food development and to promote the development of green and healthy food.

## 2. Materials and Instrumentation

### Experimental Materials

The test strains, including *W. cocos* 0014, *Ganoderma lucidum* 8, *Phellinus igniarius* MS-5 and *Trametes lactinea* CKJ, were provided by the Mycological Research Center of Fujian Agriculture and Forestry University; human breast cancer cells, MDA-MB-231, utilized as test cells, were purchased from Wuhan Pnosay Biotechnology Co. (Wuhan, China).

## 3. Experimental Methods

### 3.1. Strain Preparation

The slant strains preserved in the refrigerator were retrieved, and a small piece of the inoculating needle was used to transfer the strains in the flat medium (CYM: glucose = 35 g, yeast powder = 10 g, potassium dihydrogen phosphate = 1 g, anhydrous magnesium sulfate = 0.5 g, vitamin B1 = 0.01 g and agar powder = 3.5 g). This medium was activated for 72 h at 25 °C in the absence of light, resulting in secondary flat dish strains. Eight pieces of mycelia were obtained from the petri dish using a perforator and inserted into a liquid medium (glucose: 35 g, yeast powder: 10 g, potassium dihydrogen phosphate: 1 g, anhydrous magnesium sulfate: 0.5 g and vitamin B1: 0.01 g) for shaking flask cultivation at 28 °C and 180 r/min for 10 d. This process aimed to achieve a higher growth rate of the tertiary liquid strains for standby.

### 3.2. Microbial Source Screening

#### 3.2.1. Optimal Microbial Source Screening

In the pre-experiment, *Ganoderma lucidum*, *Phellinus igniarius* and *Trametes lactinea* strains were screened to assess their high biomass and active component levels. Mycelial growth was induced by mixing and culturing the two strains in the same liquid medium [21,22]. *W. cocos* served as the substrate and combined with *Ganoderma lucidum*, *Phellinus igniarius* and *Trametes lactinea* in pairs. During inoculation, the four strains were linked sequentially on a common substrate, initially with 5 mL of a liquid strain, and subsequently connected to another 5 mL of a different strain within 5 d. The culture was then maintained for 10 d to identify the most effective co-fermentation culture combination [23]. Based on the screening results of the co-fermentation culture combinations, the co-fermentation time underwent further evaluation. The incubation time was calculated from the initial inoculation, with intervals set at 8, 10, 12, 14 and 16 d. The screening results of the co-fermentation culture combinations served as a basis for further screening.

#### 3.2.2. Determination of Intracellular Polysaccharide Content

The intracellular polysaccharide content was determined following Zhongxue’s method with slight modifications, with anhydrous glucose utilized as the standard [24]. A glucose standard solution was prepared, and a standard curve was drawn. To achieve this, 1 mL of the prepared glucose standard solution was mixed with 1 mL of a 6% phenol solution, followed by the addition of 5 mL of concentrated sulfuric acid. The solution was allowed to stand for 15 min until cooled, then placed in a 30 °C water bath at a constant temperature of 20 min. Subsequently, a UV spectrophotometer was employed to measure the absorbance at 490 nm to draw the standard curve: Y = 0.0113x + 0.0096 R2 = 0.9987

The polysaccharide content was determined according to the phenol-sulfuric acid method [25,26], and the mycelia were collected via filtration, rinsed in distilled water for cleaning, vacuum freeze-dried and weighed. Extraction was carried out following the hydroalcohol precipitation method [27,28]. Polysaccharide precipitation was obtained and dissolved by adding 2 mL of deionized water, and the protein was removed using a Savage reagent, which was repeated twice [29]. The absorbance was measured at 490 nm using a UV spectrophotometer and recorded as A. The polysaccharide content was calculated using the following standard curve formula:Polysaccharide content (mg/g) = ((A − 0.0096)/0.0113 × 25)/100

### 3.3. Microbial Diversity in ITS Second-Generation Sequencing

Based on the results of co-culture time screening, the mycelium with the optimal culture time was sent to Beijing Prime Biotechnology Co. (Beijing, China) for detection of microbial species abundance.

### 3.4. Effects of Exogenous Substances on the Antioxidant Activity of Intracellular Polysaccharides in W. cocos Bodies

#### 3.4.1. Extraction of Intracellular Polysaccharides and Preparation of Some Reagents

The *W. cocos* intracellular polysaccharides were extracted following the method mentioned in Section 3.2.2. The polysaccharide solution was dialyzed at 100 times the volume of running water for 24 h to remove salts and other impurities in the polysaccharide solution, followed by freeze-drying to obtain crude polysaccharide samples.

#### 3.4.2. Determination of DPPH Scavenging Ability

An appropriate amount of polysaccharide sample was collected (Table 1) and prepared in a polysaccharide solution with a concentration of 5 mg/mL [30], which was subsequently diluted to concentrations of 2, 1, 0.8, 0.6, 0.4 and 0.2 mg/mL, and prepare a 0.08 mg/L DPPH solution. The experimental group, containing 2 mL of each concentration of polysaccharide solution and 2 mL of DPPH solution, was labeled as Sample A. The blank group, containing 2 mL of deionized water and 2 mL of DPPH solution, was labeled as Blank A, and the control group containing 2 mL of each concentration of polysaccharide solution and 2 mL of anhydrous ethanol was labeled as Control A [31]. The control group contained 2 mL of each polysaccharide solution and 2 mL of anhydrous ethanol. The reaction was allowed to stand for 30 min at 25 °C, and the absorbance value was measured at 517 nm via a UV spectrophotometer [32,33,34,35]. The aforementioned experiments were repeated with VC serving as a counter−positive control, and the DPPH clearance rate was calculated as follows:DPPH clearance (%) = [1 − (A sample − A control)/A blank] 100%

#### 3.4.3. Determination of Scavenging Ability for ABTS

The preparation of polysaccharide solution (Table 1) was carried out following the procedure outlined in Section 3.4.2. Preparation of ABTS+ working solution was as follows: 7 mmol/L ABTS solution and 15 mmol/L K2S2O8 solution were mixed and left to stand according to 5:1, and diluted with deionized water, and the absorbance value was 0.7 ± 0.02 at 734 nm on the visible spectrophotometer. Utilizing a clean test tube, 2 mL of each concentration of polysaccharide solution and 2 mL of ABTS + working solution were added to the experimental group, designated as A sample. Meanwhile, 2 mL of deionized water and 2 mL of ABTS + working solution were added to the blank group, designated as Blank A. Furthermore, 2 mL of each concentration of polysaccharide solution and 2 mL of deionized water were added to the control group, designated as Control A. A static reaction was allowed to occur at 25 °C for 30 min. The absorbance at 734 nm was measured using a UV spectrophotometer. The above experiments were repeated with VC serving as the counter-positive control. The formula used for calculating ABTS clearance was as follows:ABTS clearance (%) = [1 − (A sample − A control)/A blank] 100%

#### 3.4.4. Determination of Hydroxyl Radical Scavenging Ability

The polysaccharide solution (Table 1) was prepared following the procedure reported in Section 3.4.2. In a clean test tube, 0.1 mL of each concentration of polysaccharide solution, 0.5 mL of FeSO_4_ solution and 0.15 mL of sodium salicylate solution were added to the experimental group [36,37]. After thorough mixing, the mixture was supplemented with 0.35 mL of 3% H_2_O_2_ solution, designated as Sample A. Approximately 0.1 mL of deionized water, 0.5 mL of Fe_S_O_4_ solution, and 0.15 mL of sodium salicylate solution were used. After thorough mixing, the mixture was supplemented with 0.35 mL of 3% H_2_O_2_ solution and designated as Blank A. Meanwhile, 0.1 mL of each concentration of polysaccharide solution, 0.5 mL of Fe_S_O_4_ solution, 0.15 mL of sodium salicylate solution and 0.35 mL of deionized water were added to the control group. The substances were mixed well, and the resulting mixture was recorded as Control A. The reaction was allowed to stand for 30 min at a temperature of 37 °C, and the absorbance values were measured via a UV spectrophotometer at 562 nm. The above experiments were repeated with VC serving as a positive control. The hydroxyl radical scavenging rate was calculated using the following formula:Hydroxyl radical scavenging rate (%) = [1 − (Sample − Control)/Blank] 100%.

### 3.5. In Vitro Inhibition of Cell Proliferation Activity and Inhibition of Cell Migration Activity

#### 3.5.1. Cell Culture

The tumor cell line was human breast cancer cells MDA-MB-231, which was purchased from Wuhan Punosai Biotechnology Co. (Wuhan, China). The specific culture conditions were as follows: 56 mL of fetal bovine serum and 5.6 mL of penicillin-streptomycin solution in 500 mL DMEM medium, the content of penicillin was 10 KU/mL and the content of streptomycin was 10 mg/mL. Incubation was performed in an incubator at 37 °C with 5% CO_2_ content. Different concentrations of drugs were prepared and their IC50 values were calculated.

#### 3.5.2. Cell Proliferation Assay via MTT Method

The cells in the logarithmic growth phase were utilized [38,39]. The original culture medium plus PBS buffer was discarded, and the cells were washed, digested with trypsin and centrifuged to discard the supernatant at the end of digestion. An appropriate amount of cell culture medium was added, and 200 μL of cells were inoculated in each well of a 96-well plate. The plate was then incubated in a CO_2_ incubator for 24 h until adherence to the wall was achieved. Following adherence, the culture medium was aspirated and discarded. In the experimental group, the polysaccharide samples (Table 1) were prepared in polysaccharide cell culture medium with concentrations of 0.025 mg/mL, 0.05 mg/mL, 0.25 mg/mL, 0.5 mg/mL, 1 mg/mL, 2 mg/mL and 5 mg/mL. Then, 200 μL of this polysaccharide culture medium and cells were added to each well. The blank group received 200 μL of cell culture medium, while the control group received 200 μL of cell culture medium. The cells and normal culture medium were added to the blank group, and the control group was incubated for 24 h after drug administration. The cell culture medium was discarded from each well, and MTT and cell culture medium were added and incubated for 4 h [40,41]. Approximately 150 μL of dimethylsulfoxide (DMSO) was added to the culture medium and discarded. After discarding the DMSO, Metazan was dissolved by shaking the wells, and the OD value was measured at 490 nm using an enzyme counter. The absorbance values of the experimental, blank and control groups were recorded as Sample A, Blank A and Control, respectively. The cell survival rate was calculated using the following formula:Cell survival rate (%) = (Sample A − Blank A/Control A − Blank A) 100%

#### 3.5.3. Cell Migration Assay via the Cell Scratching Method

Straight lines were scratched across the wells on the back of a 6-well plate, intersecting three straight lines per well [42]. The cells in the logarithmic growth phase were washed with PBS, digested with trypsin and centrifuged to discard the supernatant. The cell culture medium was then resuspended, and each well was inoculated with 2 mL of the cells. The cells were placed in a CO_2_ incubator until the cell monolayer spread to the bottom of the wells. The center of the cell monolayer was scratched with a lance tip, the original culture medium was discarded and the cells were washed with PBS. In the experimental group, the polysaccharides (Table 1) were configured into a polysaccharide serum-free culture medium with 0.025 mg/mL, 0.05 mg/mL, 0.25 mg/mL, 0.5 mg/mL, 1 mg/mL, 2 mg/mL and 5 mg/mL, and 1.5 mL of the polysaccharide serum-free culture medium was added into each well. The scratched area was recorded as S0 by capturing images under a microscope. After 24 h of incubation in a CO_2_ incubator, the scratched area S1 was recorded. The cell migration rate was calculated as follows: Cell migration rate (%) = (S0 − S1)/S0 × 100%

## 4. Transcriptomics Analysis of Significant Differences in *W. cocos* Gene Expression Induced by Exogenous Substances

### 4.1. Sample RNA Extraction, Quality Testing and Library Sequencing

Sample RNA extraction was performed following the TRIzol method: a suitable amount of mycelium was ground, and Trizol reagent was added for a 5-min reaction at room temperature [43]. Next, 0.2 mL of chloroform was added, mixed thoroughly and left for 10 min. After centrifugation, the supernatant was collected and an equal volume of pre-cooled isopropanol was added, mixed well and centrifuged to discard the supernatant. Approximately 75% of the volume of DEPC ethanol was added to wash the precipitate. Following thorough mixing and another round of centrifugation for 5 min, the supernatant was discarded and DEPC water was added to dissolve the precipitate. The RNA purity and integrity were examined. After all the samples passed the test, 1.5 μg of RNA was collected from each sample and sent to Beijing Ovison Gene Technology Co. (Beijing, China).

### 4.2. Differential Gene Screening

According to the results of differential gene analysis, the criteria of log2FC ≥ 1 (indicating the ratio of expression between two samples, where log2FC is the base 2 logarithm of the fold change) or log2FC ≥ −1, along with a false discovery rate (FDR) of ≤0.01, were used as the screening conditions. The distribution map of the differential genes was generated using R (version 4.1.3) software.

### 4.3. GO Clustering Analysis and KEGG Enrichment Analysis

Clustering analysis and KEGG enrichment analysis were performed using the cloud platform of Ovison Gene Technology Co.

## 5. Data Processing and Analysis

SPSS (version 25.0) was used for data processing, software was used to process the scratch images and GraphPad Prism (version 9) software was used for statistical graphing. A *p* value of <0.05 indicated a significant difference, while a *p* value of <0.01 indicated a highly significant difference.

## 6. Results and Analysis

### 6.1. Microbial Source Screening

#### 6.1.1. *Ganoderma*-*Wolfiporia cocos* Co-culture

In the plate standoff test, *Wolfiporia cocos* and Ganoderma did not produce inhibition lines on a solid medium (Figure 1A), indicating the absence of antagonism.

The mycelial biomass values of *Ganoderma*-*Wolfiporia cocos* co-culture reached 7.93 g/L, 8.28 g/L, 7.17 g/L and 7.90 g/L when the inoculation method involved *Wolfiporia cocos* pre-inoculation for 2, 3, 4 and 5 d (Figure 1B1, PG0−PG5). These values were significantly higher than those of *Wolfiporia cocos* alone (5.25 g/L) and Ganoderma alone (4.33 g/L). Notably, the biomass increased when inoculated with Wolfiporia pre-inoculation for 3 d. The biomass after *Wolfiporia cocos* pre-inoculation for 3 d was significantly higher than that of Ganoderma alone (4.33 g/L). The biomass of *Wolfiporia cocos* was most significantly increased 3 d after inoculation, which was 57.71% higher than that of *Wolfiporia cocos* alone and 91.22% higher than that of *Ganoderma lucidum* alone. The intracellular polysaccharide content of the mycelia of W. lucidum was significantly higher in all five treatments compared with *Ganoderma lucidum* alone. When the inoculation method involved *Wolfiporia cocos* pre-inoculation for 3, 4 and 5 d (Figure 1B2, PG3–PG5), the intracellular polysaccharide content of mycelia was significantly elevated compared with the *Wolfiporia cocos* alone culture (192.06%, 195.37% and 209.33%) and *Ganoderma lucidum* alone culture and elevated by 44.03%, 45.67% and 47.84% compared with *Wolfiporia cocos* alone culture, respectively; no significant difference was found in the intracellular polysaccharide content of mycelia among the three inoculation methods. No significant differences were also observed between the three inoculation methods. Therefore, *Wolfiporia cocos* was pre-inoculated for 3 d and inoculated with *Ganoderma lucidum* as the inoculation type of *Ganoderma*-*Wolfiporia cocos* co-culture.

#### 6.1.2. Screening Results of *Wolfiporia cocos*-*Trametes lactinea* and *Wolfiporia cocos*-*Phellinus igniarius* Co-Culture Methods

In the plate stand-off test, an antagonistic interaction was observed between *W. cocos* and *Trametes lactinea*. When the inoculation method was used to inoculate both *Wolfiporia cocos* and *Trametes lactinea*, the biomass reached a maximum of 10.04 g/L, which was significantly higher than that of *Wolfiporia cocos* alone (6.79 g/L) and *Trametes lactinea* alone (9.06 g/L). Meanwhile, the intracellular polysaccharide content of mycelium (34.92 mg/g) was the highest, surpassing that of *W. cocos* alone (19.36 mg/g) and *Trametes lactinea* butyricum alone (27.54 mg/g). Therefore, the inoculation method used for *W. cocos*-*Trametes lactinea* co-culture was selected to inoculate both *W. cocos* and *Trametes lactinea*.

The biomass remained within the range of 8.74–9.46 g/L when the incubation time was increased from 8 d to 16 d. No significant differences were observed among these levels. The highest intracellular polysaccharide content (35.69 mg/L) was achieved after an incubation period of 8 d. The polysaccharide content significantly decreased when the incubation time was extended. Therefore, an incubation time of 8 d was selected for *Trametes lactinea*-*W. cocos* co-culture.

No antagonism was observed between *Wolfiporia cocos* and *Phellinus igniarius* in the plate standoff experiment. Inoculating *Wolfiporia cocos* for 3 d and subsequently with *Phellinus igniarius* resulted in a maximum mycelial biomass of 7.25 g/L, representing an improvement compared with that of *Wolfiporia cocos* alone (6.97 g/L); the intracellular polysaccharide content (27.72 mg/g) improved by 43.20% compared with that of *Wolfiporia cocos* alone (19.36 mg/g). When *Phellinus igniarius* was pre-inoculated for 1, 2, 3, 4 and 5 d, the biomass levels reached 8.83 g/L, 9.10 g/L, 9.13 g/L, 9.44 g/L and 9.49 g/L, respectively, which were lower than the biomass of *Phellinus igniarius* fermentation alone (9.57 g/L). Therefore, the inoculation method for the *Phellinus igniarius*-*W. cocos* involved pre-inoculating *W. cocos* for 3 d and subsequently inoculating it with *Phellinus igniarius*.

The biomass of the *Phellinus igniarius*-*W. cocos* co-culture reached a maximum of 8.90 g/L after an incubation time of 12 d. The biomass significantly decreased when the incubation time increased. The polysaccharide content reached a maximum of 34.92 mg/g after 10 d, which was significantly higher than that observed in other incubation time levels. Therefore, the incubation time of the *Phellinus igniarius*-*W. cocos* co-culture was chosen as 10 d.

### 6.2. Effect of Exogenous Substances on the Antioxidant Activity of Intracellular Polysaccharides in Wolfiporia Bodies

Table 2 presents a comparative analysis of the antioxidant activity between PG3-P and polysaccharides from Ganoderma-*W. cocos* co-fermentation mycelia. PG3-P exhibited a higher DPPH scavenging activity compared with P-P and was essentially comparable to P_G-P. PG3-P demonstrated a slight reduction in ABTS scavenging capacity compared with P-P and P_G-P. The scavenging capacity of hydroxyl radicals was stronger than that of both P-P and P_G-P.

In the experiments assessing the antioxidant capacity of the mycelial polysaccharides of *Trametes lactinea*-*W. cocos* co-fermentation, the scavenging capacity of PT0-P for DPPH was higher compared with that of both P-P and P_T-P. However, for the scavenging capacity of ABTS and hydroxyl radicals, the scavenging activity of PT0-P weakened to some extent compared with that of both P-P and P_T-P.

Illustrates a comparative analysis of the antioxidant capacity of the mycelial polysaccharides of *Phellinus igniarius*-*W. cocos* co-fermentation and PPi3-P. Results revealed that PPi3-P exhibited higher activity in scavenging DPPH and hydroxyl radicals compared with P-P and P_ Pi-P. With regard to the scavenging capacity of ABTS, the scavenging activity of P Pi3-P was stronger than that of P_Pi-P and was essentially the same as that of P-P.

### 6.3. Evaluation of Tumor Cell Proliferation-Inhibition Activity

Figure 2 illustrates the results of *Ganoderma lucidum*-*W. cocos* co-fermented mycelial polysaccharide inhibition of breast cancer cell activity in vitro. Under the same dosage conditions, PG3-P (IC50 = 1.756 mg/mL) demonstrated a significantly higher inhibition of breast cancer cell activity in vitro compared with P_ G-P. The exception was at a concentration of 5 mg/mL treatment, which showed no significant difference from P-P. However, in other concentrations, a highly significant difference was observed. Within the concentration range of 1–5 mg/mL, the cell survival rate remained between 44.56% and 47.21%, with no significant difference between the cell survival rates of the three concentration gradient treatments. These findings indicate that PG3-P has stronger activity than both P-P and P_G-P in inhibiting breast cancer cells in vitro.

The antiproliferative activity of PT0-P was lower than that of both P_T-P and P-P, while its IC50 values were higher than those of the latter two. This observation suggests that the antitumor proliferative activity of *Wolfiporia cocos* polysaccharides could not be enhanced by induction with the microbial source, *Trametes lactinea*.

The antiproliferative activity of PPi3-P was lower than that of both P_Pi-P and P-P, and its IC50 values were higher than those of the latter two. This finding indicates that the antitumor proliferative activity of the polysaccharides could not be enhanced by induction with the microbial source, *Phellinus igniarius* (Table 3).

### 6.4. In Vitro Evaluation of Breast Cancer Cell Migration-Inhibition Activity

At a concentration of 0.05 mg/mL, the migration rate of cells incubated with PG3-P was significantly different from that of the control group. By contrast, the concentration of P_G-P had to reach 0.25 mg/mL before the cell migration rate reached a significant level (Figure 3). As the polysaccharide concentration increased, cell mobility decreased and the lowest mobility was observed at a concentration of 5 mg/mL, with migration rates of 28.34% and 37.62% for PG3-P-and P_ G-P-treated cells, respectively. This finding implies that the in vitro inhibition of breast cancer cell migration by *Ganoderma lucidum*-*Wolfiporia cocos* co-culture mycelial polysaccharide is more potent than that by *Ganoderma lucidum* and *W. cocos* alone fermented mycelial polysaccharide.

Compared with *Trametes lactinea*-*W. cocos* co-fermented mycelia, the antitumor cell migration activity of polysaccharides from *Trametes lactinea* and *W. cocos* alone was weaker. Even with an increase in PT0-P concentration, the migration rate of breast cancer cells did not differ significantly from that of the blank group. By contrast, the concentration of P_T-P reached 0.025 mg/mL, while that of P-P reached 1 mg/mL, which showed significant differences from those of the control group. Therefore, the induction by *Trametes lactinea* failed to increase the antitumor cell migration activity of polysaccharides from *W. cocos*. This indicates that induction by *Trametes lactinea* failed to improve the antitumor cell migration activity of *Wolfiporia cocos* mycelial polysaccharides.

Similarly, the antitumor cell migration activity of polysaccharides from *Phellinus igniarius*-*Wolfiporia* co-fermented mycelia was weaker than that of *Phellinus igniarius* and *Wolfiporia cocos* co-fermented mycelia alone. At a concentration of 5 mg/mL, the migration rate of PPi3-P-treated breast cancer was 39.11%, while that of P_Pi-P-treated breast cancer was 16.40%. This observation implies that *Phellinus igniarius* induction failed to enhance the antitumor cell migration of polysaccharides from *W. cocos* mycelia.

#### 6.4.1. Transcriptome Sequencing Quality Analysis

The transcriptome data of each group of mycelium samples exhibited clean reads/raw reads exceeding 98%, a data error rate of ≤0.03% and base quality control Q20 rate of ≥97.80% and Q30 rate of ≥93.81%. These metrics indicate that the transcriptome assembly was performed using high-quality data and under reliable conditions, facilitating further analysis for research purposes.

#### 6.4.2. Inter-Sample Correlation Analysis

Gene expression shows biological variability among individuals, and different genes show varying degrees of expression variability. To identify differentially expressed genes (DEGs), the expression differences caused by biological variability need to be considered. The assessment of the correlation among biological duplicates is essential for analyzing the transcriptome sequencing data. Spearman’s correlation coefficient R.R^2^ was used as an index to evaluate the correlation of biological repeats. The R^2^ was closer to 1, indicating a stronger correlation between the two repeats. The gene expression correlation R^2^ between the samples was ≥0.956 (Figure 4), signifying excellent duplication within each group of PG3, PT0 and PPi3 samples. This allows for detailed analysis and evaluation of samples.

### 6.5. Screening of Differentially Expressed Genes

The differences in gene expression levels between the samples were analyzed using a volcano plot (Figure 5). Each scatter represents a gene, with the vertical axis denoting the significance of differential gene expression between the groups and the horizontal axis indicating the multiplicity of DEGs in the samples. When the PG3 group was compared with the P group, 64 DEGs were identified, of which 27 were upregulated and 37 were downregulated. Conversely, when the PPi3 group was compared with the P group, 524 DEGs were identified, of which 239 were upregulated and 285 were downregulated.

### 6.6. GO Enrichment Analysis of Differentially Expressed Genes

Gene Ontology (GO) analysis categorizes differential genes according to biological processes, cellular components and molecular functions, and determines the functions specific to the differential genes. The GO analysis of DEGs in the PPi3vsP group revealed enrichment in the biological processes such as oxidation reduction and lipid metabolism. Molecular functions, including oxidoreductase activity, oxidoreductase activity, lipid metabolism and lipid metabolism, were also enriched, with a particular emphasis on the oxidoreductase and catalytic activities (Figure 6).

### 6.7. KEGG Enrichment Analysis of Differentially Expressed Genes

KEGG enrichment analysis of DEGs was performed, and the 40 most significantly differentially enriched pathways were selected for KEGG bubble mapping. The primary enriched metabolic pathways were fatty acid metabolism, endocytosis, amino acid biosynthesis and carbon metabolism (Figure 7).

To explore the mechanism by which exogenous substances upregulate the intracellular polysaccharide content of *W. cocos*, we analyzed the DEGs shared by the PG3 group vs. P group and the PPi3 group vs. P group. The results of this analysis were combined with the KEGG database to compare the enriched pathways of the DEGs. Our findings revealed participation in the biosynthesis of N-glycans (Figure 8). In the entire metabolic pathway, ALG10 was upregulated, while MAN1B was downregulated. We hypothesized that exogenous substances promote the processing and modification of glycan precursors by upregulating the expression of α-glucosyltransferase encoded by ALG10. This in turn promotes the synthesis of polysaccharides. Conversely, the metabolism of glycan catabolism downregulates the expression of α-glycosidylglucosidase encoded by MAN1B, subsequently inhibiting its metabolism. Consequently, the polysaccharide content in *W. cocos* can be enhanced.

## 7. Conclusions

At present, most researchers use multiple means to increase the polysaccharide content in liquid fermentation, while co-fermentation is a technology that utilizes fungal or bacterial fermentation of dual-use substrates, which has synergistic and complementary characteristics that can increase the yield of the active ingredient and can be introduced to a certain extent into the active ingredient, which enhances its pharmacological effects and clinical applications. In order to investigate the mechanism of upregulation of *Wolfiporia cocos* intracellular polysaccharides by exogenous microorganisms, transcriptomics was used to study *Wolfiporia cocos* mycelium after induction by exogenous microorganisms, and finally an enrichment pathway, i.e., N-glycan metabolism pathway, was found to be related to polysaccharide metabolism, and it was hypothesized that exogenous substances could upregulate the expression of α-glucosyltransferase, which is encoded by ALG10, and downregulate α-glucuronidases, which is encoded by MAN1B, in the catabolic metabolism of polyglycans. α-glucosidase expression. Overall, for the polysaccharide yields obtained via liquid co-fermentation of *Wolfiporia cocos*, there was a great improvement over the traditional way. These polysaccharides still have good antioxidant and antitumor abilities as verified by physiological functions; this again demonstrates the value of liquid co-fermentation of *Wolfiporia cocos*, and we can optimize this co-fermentation system again and apply it to other foods or traditional Chinese medicines with health care functions to promote the development of functional foods.

## Figures and Tables

**Figure 1 foods-13-01578-f001:**
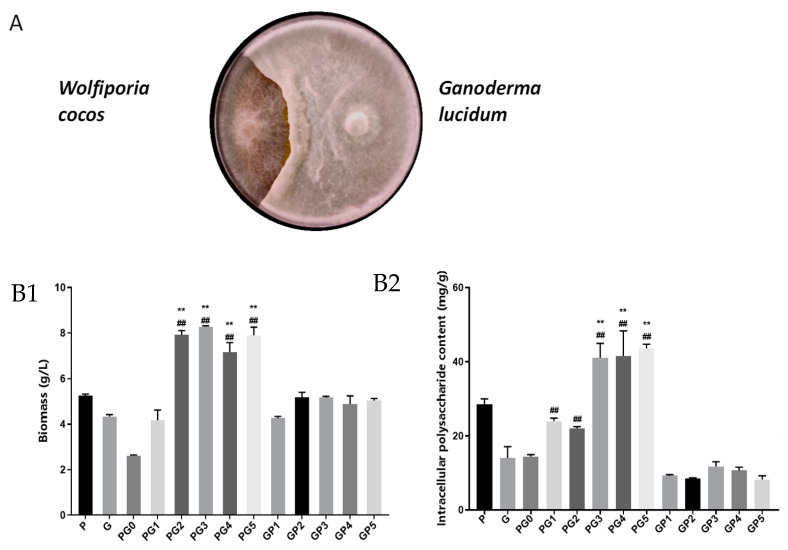
Effects of different inoculation methods on mycelium biomass and intracellular polysaccharide in co-culture of *W. cocos* and *G. lucidum*. Note: contrast with *W. cocos* by fermentation alone, ** *p* < 0.01; contrast with *G. lucidum* by fermentation alone, ## *p* < 0.01; (**A**) confrontation between *G. lucidum* and *W. coco*; (**C**) mycelium growth of different inoculation methods. (**B1**) is the mycelial biomass of different co-fermentation systems; (**B2**) is the intracellular polysaccharide content of mycelium in different co-fermentation systems.

**Figure 2 foods-13-01578-f002:**
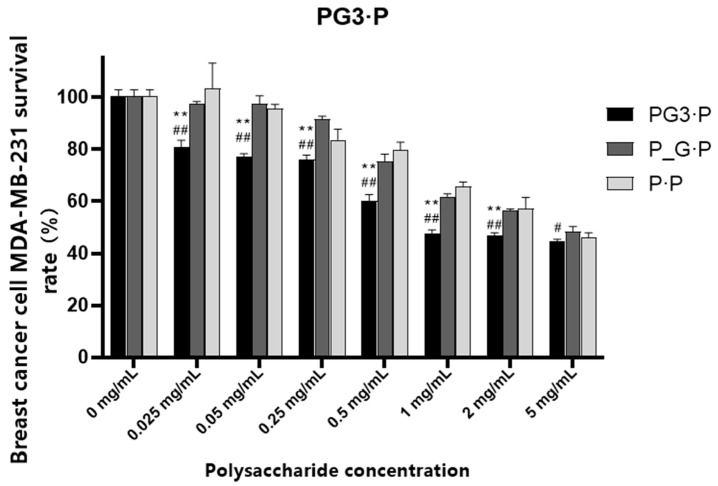
Effect of polysaccharide from the mycelia of *W. cocos* induced by *G. lucidum* on the growth of breast cancer cells. Note: at the same concentration, P·P treatment decreased significantly, ** *p* < 0.01; at the same concentration, P_G·P treatment decreased significantly, # *p* < 0.05, ## *p* < 0.01.

**Figure 3 foods-13-01578-f003:**
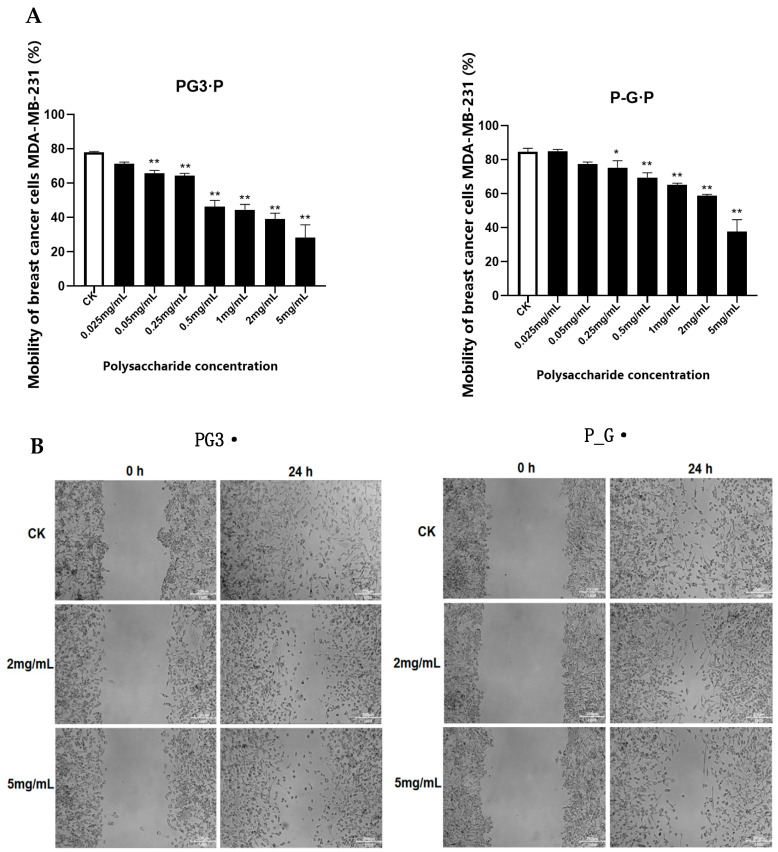
(**A**) Effect of *G. lucida*-*W. cocos* fermented mycelium polysaccharide on mobility of breast cancer cells. Note: as the polysaccharide concentration increased, the migration rate was significantly reduced in comparison with the control, * *p* < 0.05, ** *p* < 0.01; (**B**) comparison of cell scratches before and after polysaccharide treatment.

**Figure 4 foods-13-01578-f004:**
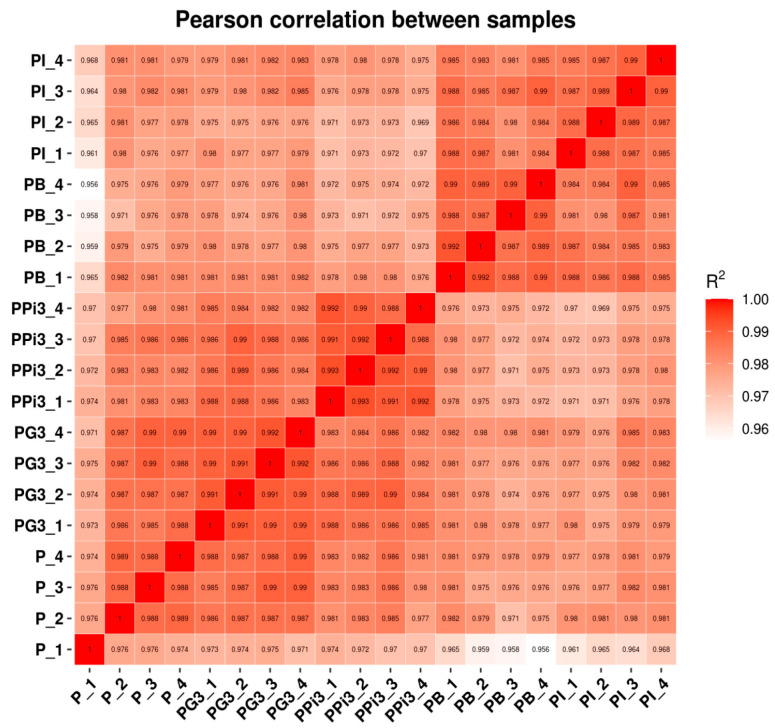
The R2 values of the 12 sets of data p_1 to ppi3_4 are all close to 1. Correlationanalysis of transcriptome sequencing data. Spearman’s correlation coefficient R was used as an assessment of biological replicate correlation. The closer R^2^ is to 1, the stronger the correlation between two replicate samples.

**Figure 5 foods-13-01578-f005:**
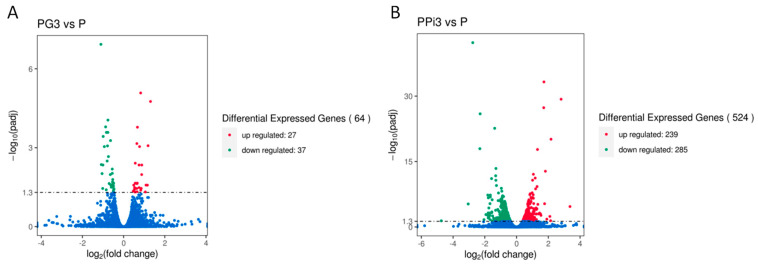
Differential gene volcano map. Note: red dots indicate gene upregulation; green dots indicate gene downregulation; (**A**) PG3vsP; (**B**) PPi3vsP.

**Figure 6 foods-13-01578-f006:**
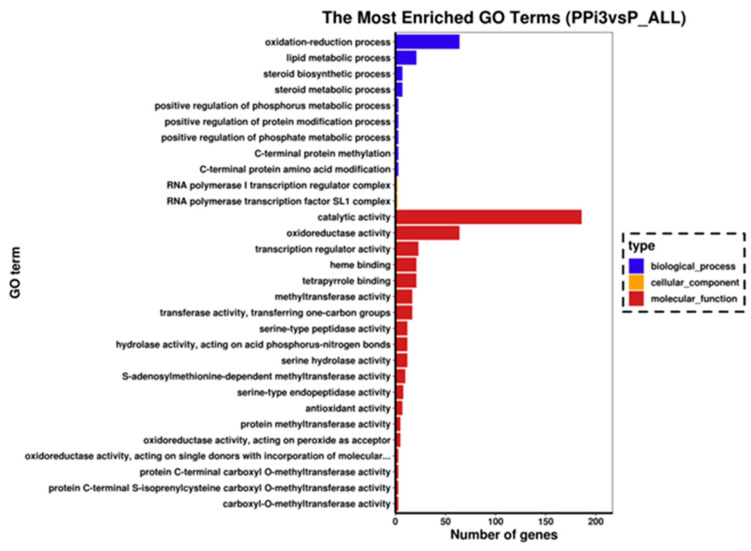
GO enrichment analysis of DEGs in PPi3vsP.

**Figure 7 foods-13-01578-f007:**
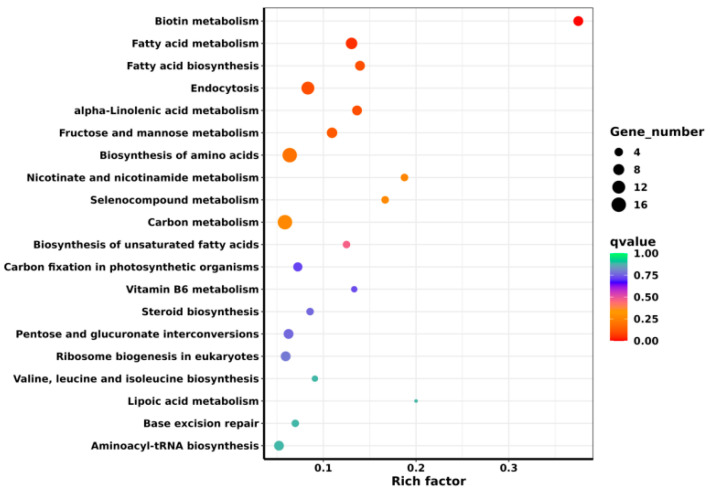
Bubble chart of differential gene KEGG enrichment.

**Figure 8 foods-13-01578-f008:**
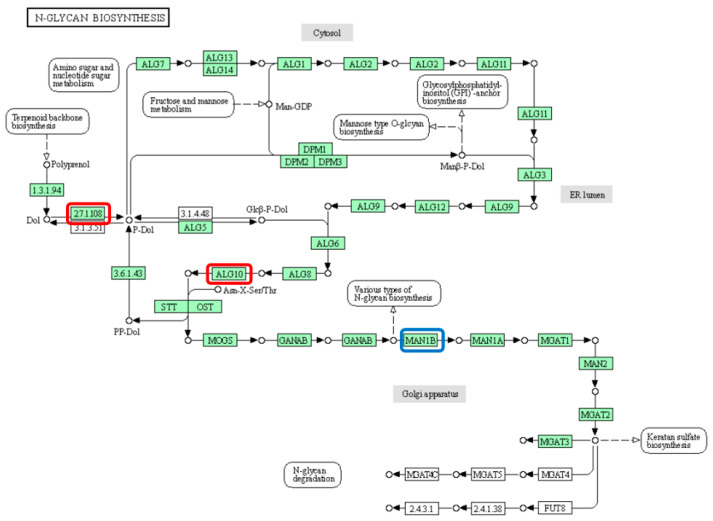
Diagram of the N-glycan metabolic pathway. Note: the red box indicates an upward adjustment; blue boxes indicate downward adjustments.

**Table 1 foods-13-01578-t001:** Various polysaccharides and sources.

Name	Sources
P·P	Mycelial polysaccharides from *Wolfiporia cocos* fermentation without the addition of any exogenous inducing substances
G·P	Mycelial polysaccharides from *Ganoderma lucidum* fermentation without adding any exogenous inducing substances
T·P	Mycelial polysaccharides from the fermentation of *Trametes lactinea* without the addition of any exogenous inducing substances
Pi·P	Mycelial polysaccharides from *Phellinus igniarius* fermentation without the addition of any exogenous inducing substances
PG3·P	Mycelial polysaccharides from co-cultures of *Ganoderma lucidum* and *Wolfiporia cocos*
PT0·P	Mycelial polysaccharides from co-cultures of *Trametes lactinea* and *Wolfiporia cocos*
PPi3·P	Mycelial polysaccharides from co-cultures of *Phellinus igniarius* and *Wolfiporia cocos*
P_G·P	Mixed mycelial polysaccharides from separate fermentations of *Ganoderma lucidum* and *Wolfiporia cocos* in the ratio of 1:1
P_T·P	Mix of mycelial polysaccharides from separate fermentations of *Trametes lactinea*, *Wolfiporia cocos* in a ratio of 1:1
P_Pi·P	Mixed mycelial polysaccharides from separate fermentations of *Phellinus igniarius* and *Wolfiporia cocos* in the ratio of 1:1

**Table 2 foods-13-01578-t002:** Comparison of free radical scavenging ability.

Polysaccharide Name	EC_50_ (mg/mL)		
	DPPH	ABTS	Hydroxyl Radical
PG3·P	0.118	0.111	0.125
P·P	0.18	0.074	0.165
P_G·P	0.116	0.097	0.284
PT0·P	0.161	0.163	0.142
P·P	0.18	0.074	0.165
P_T·P	0.178	0.133	0.152
PPi3·P	0.076	0.077	0.101
P·P	0.18	0.074	0.165
P_Pi·P	0.203	0.227	0.203

**Table 3 foods-13-01578-t003:** Comparison of the proliferation inhibitory activity of polysaccharides on breast cancer cells.

Polysaccharide Name	IC_50_ (mg/mL)
PT0·P	7.194
P_T·P	3.519
P·P	3.318
PPi3·P	7.701
P_Pi·P	4.020
P·P	3.318

## Data Availability

The original contributions presented in the study are included in the article, further inquiries can be directed to the corresponding author.

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
