# Peer review of "Mechanistic and Functional Studies on the Microbial Induction of Wolfiporia cocos Liquid Fermentation Products"

_foods, 2024, doi:10.3390/foods13101578_

Round 1

Reviewer 1 Report

Comments and Suggestions for Authors

The objective of the current study was to investigate the changes in the total polysaccharide content and bioactivity through the introduction of exogenous microbes to the three different Wolfiporia cocos liquid cofermentation systems. The topic is very interesting and the manuscript needs major modifications below before publication

Authors should follow the journal instructions mainly in references citation and format

Abstract: at the end of abstract, authors should show the importance of theirs results and their practical applications

Introduction, L78: the significance of exogenous substances in the fermentation system should be elaborated and more discussed in the introduction with citation some related references.

Methodology L105-106: please add reference to this sentence? Has this mothds applied in previous studies and how you make sure that the growth rates of microbes are the greatest?

L151: this is table not figure please reorganize

The section of results should be summarized and write the most important results not to include all numerical data in the figures and tables.

Figures and Tables should be correctly cited in the text.

L342: this is table not figure, the same L349, 357, 379, and 385. These should be combined in one table

Figures should be re-numbered. All figures are numbered as “Fig. 6”

Discussion: the practical application of the experiment results and their significance should be discussed.

Comments on the Quality of English Language

Minor editing of English language required

Author Response

Dear reviewer

Thank you very much for your careful reading of my manuscript, in which errors have been corrected or deleted accordingly and highlighted in blue in the text. I have revised your questions in the form of questions and answers for the benefit of the reviewers. At the same time, if there is any unreasonable place in the manuscript, please point it out in time, and I will revise it as soon as possible.

I wish you a happy life.

Kind regards,

Zhikang Yang

Reviewer 2 Report

Comments and Suggestions for Authors

I have revised one more time the document Foods-2977515. Unfortunately, the document is very difficult to follow: (1) not all figures are referenced in text; thus, their purpose is unknown, (2) figures cannot be understood on its own as the caption does not provide enough information about results shown and/or legends (the required information is not included in text), (3) several values are presented without a statistical analysis and a measure of experimental error, (4) the way in which some statistical analysis is presented is not adequate and the figure captions do not provide clear information about what was done. Thus, I have included additional comments about the study in this e-mail. Best regards.

  1. Figure 3-1. This figure is not referenced in text.
  2. Section 6.1.1. (a) Lines 265-268. The inoculation times were not clearly defined in methodology. (b) Figure 6-1. Legends were not defined. What is P, G, PG0, etc.? The authors should clearly indicate the meaning of these legends in text before its use.
  3. Line 262. “cocos cocos”.
  4. Lines 294-295, 309-310. The authors should have included an image as that shown in Figure 6-1A.
  5. I do not see the purpose of Figure 6-1C. This figure is never discussed in text.
  6. Results in Figure 6-1B should be presented as a table. This way the authors could also add results from section 6.1.2. Besides, all treatments pairs should be compared at a unique significance level (0.05) and letters could be added to indicate significant differences between them.
  7. Figure 6-2. Again, legends were not defined. What is the meaning of the numbers -1, -2, -3 after PG3, PPi3, PT0.
  8. Figures 6-3 to 6-5, 6-7, 6-8. Values lack standard deviation or other error estimate. Statistical analysis? The authors are discussing values without a proper statistical analysis.
  9. Figure 6-6. All treatments pairs should be compared at a unique significance level (0.05) and letters could be added to indicate significant differences between them.
  10. Figure 6-9. The top figure lacks the legend A. What is CK? “Figure A is significantly lower than that of the control group” What do you mean by this text? It is not clear what do symbols on bars indicate. The authors should use letters to indicate treatments differences.
  11. Figure 6-10. Numbers in figure are illegible. What is the meaning of legends? What responses are you correlating?
  12. Figure 6-14. This figure is presented without a proper explanation.
  13. Discussion is very poor. Practically, all results are presented without a proper comparison to related studies.

Author Response

(The authors gave the same response as above.)

Reviewer 3 Report

Comments and Suggestions for Authors

The manuscript 'Anticancer ability and mechanism of Poria cocos fermentation products induced by microorganisms' by Yang et al. is a good study depicting changes in polysaccharide content and bioactivity as a result of the addition of exogenous microorganisms to the Wolfiporia coconut liquid co-digestion system, but nevertheless needs some corrections and clarifications.

Below are some comments on the manuscript:

The title of the paper mentions a mechanism that is not investigated in this work.

The aim of the work should be reworded. In the back of the paper there is mention of anticancer activity, a lot of space is also devoted to this issue in the description of results, discussion and conclusions. In contrast, there is no mention of this in the objective of the work.

The abstract of the paper and the introduction are too extensive. I would ask you to shorten these two contents, to include the most important information. This particularly concerns the abstract, the volume of which does not meet the requirements of the journal

Chapter 3.4.2.Please give the concentration/molality of the DPPH solution

Section 3.4.3.Please give the concentration/molality of the ABTS solution

Chapter 3.5.1.No indication of which cancer line was studied. Such information can only be obtained from the results of the work. Please provide information on the cell culture, where the cell line was obtained from, under what conditions it was cultured. Please also provide information on how the IC50 value was calculated.

In how many replicates was the antioxidant activity determined? Why was no statistical analysis performed for this assay? 

I do not really understand the figure designations e.g. "figure 6-1", figure 6-2". Please keep the order throughout the text 'figure 1', 'figure 2', etc. and then if necessary explain the references in figures 'A', 'B', etc.

In addition, the graph "Breast cancer survival" has been repeated twice.

Description under Figure 6-9 "Figure 6-9. Effect of G.lucida-W.cocos fermented mycelium polysaccharide on mobility of breast 399 cancer cells. Note: Figure A is significantly lower than that of the control group,...", it is not stated where figure A is located. Furthermore, the methodology states that the cells in this assay were treated with 7 concentrations of polysaccharide. Microscope images are only shown for 3 concentrations. Please provide supplementary material for the other photos

Migration graph of PG3P treated cells is missing.

In conclusion, please indicate possible future research/use of the demonstrated novelty of the product.

Please adapt the literature list to the requirements of the journal.

Author Response

(The authors gave the same response as above.)

Round 2

Reviewer 1 Report

Comments and Suggestions for Authors

The manuscript is improved and can be accepted in the current form

Author Response

Dear Reviewer.
This is the first manuscript out of my submission, and there were a lot of low-level errors. Like the repetition of vocabulary and discontinuity of graphs and charts pointed out by the three reviewers. I appreciate your careful checking, and at the same time, it gives me lessons to learn. All should be carefully checked after the writing is completed.
I wish you a happy life.

Reviewer 2 Report

Comments and Suggestions for Authors

   Some figures are not discussed in text (such as Figure 1 C). Thus, their purpose is unknown.

2.       Some results (Tables 2,5, and 6) are presented without standard deviation or other error estimate, and without a proper statistical analysis (for example, Tukey’s test). I think the authors’ response is unacceptable (“The raw data will be collected as soon as possible and re-graphed”), specially as the authors dare to qualify values with terms such as “higher”, “comparable”, and “slight reduction” in the discussion.

3.       Some figure captions (for example, Figures 5 and 9) do not provide the necessary information to fully understand what is presented.

Minor comments:

Table numbering is not consecutive.

Author Response

(The authors gave the same response as above.)
